# Bacterial Community Characteristics Shaped by Artificial Environmental PM2.5 Control in Intensive Broiler Houses

**DOI:** 10.3390/ijerph20010723

**Published:** 2022-12-30

**Authors:** Wenxing Wang, Guoqi Dang, Imran Khan, Xiaobin Ye, Lei Liu, Ruqing Zhong, Liang Chen, Teng Ma, Hongfu Zhang

**Affiliations:** 1State Key Laboratory of Animal Nutrition, Institute of Animal Sciences, Chinese Academy of Agricultural Sciences, Beijing 100193, China; 2Bureau of Agriculture and Rural Affairs of Luanping County, Chengde 068250, China

**Keywords:** airborne bacteria, PM2.5, environmental control variables, manurial bacteria, broiler rearing

## Abstract

Multilayer cage-houses for broiler rearing have been widely used in intensive Chinese farming in the last decade. This study investigated the characteristics and influencing factors of bacterial communities in the PM2.5 of broiler cage-houses. The PM2.5 samples and environmental variables were collected inside and outside of three parallel broiler houses at the early, middle, and late rearing stages; broiler manure was also gathered simultaneously. The bacterial 16S rRNA sequencing results indicated that indoor bacterial communities were different from the outdoor atmosphere and manure. Furthermore, the variations in airborne bacterial composition and structure were highly influenced by the environmental control variables at different growth stages. The db-RDA results showed that temperature and wind speed, which were artificially modified according to managing the needs for broiler growth, were the main factors affecting the diversity of dominant taxa. Indoor airborne and manurial samples shared numerous common genera, which contained high abundances of manure-origin bacteria. Additionally, the airborne bacterial community tended to stabilize in the middle and late stages, but the population of potentially pathogenic bacteria grew gradually. Overall, this study enhances the understanding of airborne bacteria variations and highlighted the potential role of environmental control measures in intensive farming.

## 1. Introduction

Bioaerosols released from modern intensive large-scale and high-density broiler farming have increasingly become an environmental concern. The harmful gases, particulate matter (PM), and microorganism emissions produced in the broiler farming process cause environmental pollution and health threats to animals and workers [1]. Improving the farming environment can help to promote the production performance of livestock and poultry, reduce the occurrence of disease, and also reduce pollution released to the surrounding environment [2].

In intensive farming, airborne particulate matter is derived from the aerosolization of animal manure, feedstuff, skin, and feather fragments, which escape into the air to form bioaerosols containing a large number of bacteria. Fine particulate matter (PM2.5), which has an aerodynamic diameter of less than 2.5 μm, could be deposited in the bronchi and lungs due to its small size, causing respiratory inflammation and diseases [3]. In addition, PM2.5 has the characteristics of long-term suspension, long transmission distance, and carrying a large number of bacteria; thus, it plays an important role in air pollution and pathogen transmission [2].

Different from the atmospheric environment, PM2.5 in the broiler house contained a large number of microorganisms, most of which were bacteria. High concentrations of airborne bacteria in enclosed houses with large numbers of broilers have attracted much attention in recent years, especially regarding the characteristics and distribution of these bacteria. Skóra et al. found that the concentration of bacteria in particulate matter in poultry aquaculture house was 3.2 × 10^9^ CFU/g and the fungus concentration was 1.2 × 10^6^ CFU/g [4]. O’Brien et al. explored the bacteria in PM2.5 and TSP in chicken houses and found that *Staphylococcus* and *Salinicoccus Carnicancri* were the main bacteria in the two particles [5]. Yang et al. observed a high abundance of pathogenic bacteria and fungi such as *Escherichia*, *Corynebacterium*, *Aspergillus*, and *Penicillium* in broiler houses [6]. It was reported that Gram-positive bacteria accounted for approximately 90% of airborne bacteria [7]. The most common Gram-positive bacteria are *Staphylococcus*, *Streptococcus*, and *Enterococcus*. Gram-negative bacteria accounted for a small proportion, with common species including *Enterobacteriaceae*, *Pseudomonas*, and *Neisseria* [8]. In addition, secondary metabolites can be detected in particulate matter, including aurofusarin, deoxynivalenol, and volatile odorous compound such as ammonia and acrolein [4].

Airborne bacterial concentrations are influenced by a variety of factors, such as building design, temperature, poultry condition, manure management, and ventilation, of which ventilation is considered to be the most important factor in regulating the farmhouse environment and airborne bacterial concentrations [9]. At the same time, some researchers also found that the poultry production system (flat housing system or cage housing system) was important [10,11]. Wu et al. analyzed the influence of environmental factors on airborne bacteria and found that humidity was most closely related to the distribution of bacteria in particulate matter and had a great influence on them [12]. There were significant differences in microbial diversity and composition in poultry houses during different growth periods, which may be caused by the different growth stages of broilers [13]. High stocking density, high humidity, and high temperature lead to decreases in air quality, which is conducive to the growth of *Escherichia coli* [14]. However, there is still a knowledge gap regarding the influence of environmental factors on the distribution of airborne bacteria in broiler growth cycles.

In modern society, with a focus on green development, people are paying more and more attention to the air quality of animal feeding operations [15]. Studies of the spatial and temporal distribution of bacteria in the air of these chicken houses will guarantee to reduce the production and emission of air pollutants and create a good environment for both broilers and practitioner. Bacterial aerosols are small in size and often contain zoonotic pathogens such as *Escherichia coli* [16]. Long-term exposure to high concentrations of bacterial aerosols in the chicken house significantly reduced the immune function of broilers, slowed down weight gain, and increased the morbidity and mortality of livestock and poultry [17]. Indoor high concentrations of airborne bacteria can overload and degrade the immune system, increasing the likelihood to directly lead to respiratory diseases, resulting in losses to poultry production. Exposure to high concentrations of microbial aerosols has been associated with worsening worker health [18]. According to one report, poultry factory workers have a higher prevalence of work-related respiratory and skin diseases than other agricultural workers [19]. In order to reduce airborne particulate matters and airborne bacterial emissions, it is necessary to implement science-based control strategies and monitoring procedures for poultry producers [20].

Airborne bacterial composition and distribution are influenced by a variety of factors, such as building design, temperature, poultry condition, manure management, and ventilation, of which ventilation is considered to be the most important factor in regulating the farmhouse environment and airborne bacterial concentrations [9]. At the same time, some researchers also found that the poultry production system (flat housing system or cage housing system) was important [10,11]. Wu et al. analyzed the influence of environmental factors on airborne bacteria and found that humidity was most closely related to the distribution of bacteria in particulate matter and had a great influence on them [12]. There were significant differences in microbial diversity and composition in poultry houses during different growth periods, which may be caused by the different growth stages of broilers [13]. High stocking density, high humidity, and high temperature lead to decreases in air quality, which is conducive to the growth of *Escherichia coli* [14]. However, there is still a knowledge gap regarding the influence of environmental factors on the distribution of airborne bacteria in broiler growth cycles.

This study was conducted from June to August 2020 in an intensive broiler farm with an automated three-layer cages system. Environmental parameters of the whole culture cycle were determined, and PM2.5 samples of early, middle, and late growth stages, as well as corresponding manurial samples, were collected. Airborne and manurial bacterial communities were analyzed by 16S rRNA high-throughput sequencing. The purpose of this study was (1) to clarify the influence of environmental conditions on bacterial aerosol characteristics in intensive broiler farming, and (2) to provide a feasible reference for the pollution control of bacterial aerosols in broiler houses, so as to improve the control measurements of intensive farming and the welfare of livestock and poultry.

## 2. Materials and Methods

### 2.1. Broiler House Structure and Daily Management

This study was conducted from June to August in Luanping, Hebei Provence, China (117°33′ E, 40°94′ N). The broiler houses were positioned in an east–west orientation, with a length of 90 m, a width of 18 m, and a height of 3.8 m, covering an area of approximately 1620 m^2^. The houses were adopted the fully enclosed 3-overlap caged rearing system. The environment inside in summer relied on the automatic control system, with 4 draught fans, 13 pairs of side windows, and 1 wet curtain device controlled by negative pressure ventilation. The automatic manure belts located at the bottom of each layer of cage were operated every two days in the early stage and once a day in the middle and late stages. The Arbor Acres (AA) boiler scale was 39,000 in each house, with a 42-day feeding period. Feed and water were supplied ad libitum throughout the entire experiment. The house temperature was gradually decreased from 34 °C (D1) to 28 °C (D28). After D28 of age, the temperature was kept at 24–28 °C until the end of the experiment. The indoor sampling points were set at the end of the cages, in the center of the house, and one outdoor point was set beside the air entrance to collect the atmospheric sample. A structural sketch and sampling points of the broiler houses are shown in Figure 1.

### 2.2. Sample Collection

The PM2.5 samples and environmental control variable data were collected inside and outside the broiler house (Figure 1). A 2030-type aerosol collector (Qingdao Laoying Environment Protection Technology Co., Ltd., Qingdao, China) was used to collect PM2.5 samples on 90 mm diameter glass fiber filters. The collector was installed at a distance 1.5 m above the ground with flow rate of 100 L/min. Each collector started the 48 h sample collection at the same time: 9:00. The days of sample collection were D3, D5, D9, D13, D17, D21, D23, D25, D28, D33, D37, and D41. In total, 48 PM2.5 samples were collected, including 36 samples inside and 12 samples outside the broiler houses. The filters were sterilized at 500 °C for 3 h and maintained at moisture equilibration in an ambient atmosphere for 48 h. The mass of PM2.5 was obtained by subtracting the constant weight (W0) from the average weight after sampling (W1) of the filter. The PM2.5 concentration was calculated from the mass of PM2.5 and the total air volume filtered [21]. From D3 of age, fresh uncontaminated manure samples were collected from the manurial belt near the PM2.5 collectors every day. Every time the manure was collected, the manure of three experimental broiler houses was mixed into a cryopreservation tube and stored at −80 °C for subsequent experiments. In total, 40 tubes of manure samples were collected. The measuring ranges of the temperature and humidity meter were −20 °C to 50 °C and 0–99%, respectively, and the minimum values were 0.1 °C and 0.1%, respectively. A portable gas detector was used to measure CO_2_ and NH_3_ concentrations, with ranges of 0–5000 ppm and 0–50 ppm and minimum values of 1 ppm and 0.1 ppm, respectively.

### 2.3. DNA Extraction and PCR Amplification

According to the manufacturer’s instructions, a PowerSoil DNA Isolation Kit (MoBio Laboratories, Carlsbad, CA, USA) was used to extract DNA from the PM2.5 and manure samples. A NanoDrop 2000 UV–vis spectrophotometer (Thermo Scientific, Wilmington, NC, USA) was used to determine the concentration and purity of DNA extracts, which were checked on 1% agarose gel. The hypervariable regions V3–V4 of the bacterial 16S rRNA genes were amplified with primer pairs 338F (5′-ACTCCTACGGGAGGCAGCAG-3′) and 806R (5′-GGACTACHVGGGTWTCTAAT-3′), using an ABI GeneAmp^®^ 9700 PCR thermocycler (ABI, Foster City, CA, USA). The 16S rRNA gene amplification was performed according to the following PCR process: initial denaturation at 95 °C for 3 min, followed by 27 cycles of denaturing at 95 °C for 30 s, annealing at 55 °C for 30 s and extension at 72 °C for 45 s, a single extension at 72 °C for 10 min, and end at 4 °C. The mixtures used for the PCR amplification included the following contents: 4 μL 5× TransStart FastPfu buffer, 2 μL 2.5 mM dNTPs, 0.8 μL forward primer (5 μM), 0.8 μL reverse primer (5 μM), 0.4 μL TransStart FastPfu DNA Polymerase, 10 ng template DNA, and finally, ddH_2_O up to 20 μL. The PCRs of each sample were performed by setting three repetitions. After the PCR products were extracted from 2% agarose gel, they were purified with an AxyPrep DNA Gel Extraction Kit (Axygen Biosciences, Union City, CA, USA). A Quantus™ Fluorometer (Promega, Madison, WI, USA) was used for the quantitative analysis of PCR products [10].

### 2.4. Illumina MiSeq Sequencing

Purified amplicons were pooled in equimolar, and paired-end sequenced (2 × 300) on an Illumina MiSeq platform (Illumina, San Diego, CA, USA) according to the standard protocols by Majorbio Bio-Pharm Technology Co., Ltd. (Shanghai, China). The raw reads were deposited into the NCBI Sequence Read Archive (SRA) database.

### 2.5. Processing of Sequencing Data

The raw 16S rRNA gene sequencing reads were demultiplexed, quality-filtered by Trimmomatic, and merged by FLASH with the following criteria: (1) The 300 bp reads were truncated at any site receiving an average quality score of <20 over a 50 bp sliding window, the truncated reads shorter than 50 bp were discarded, and reads containing ambiguous characters were also discarded; (2) Only overlapping sequences longer than 10 bp were assembled according to their overlapped sequence. The maximum mismatch ratio of overlap region was 0.2. Reads that could not be assembled were discarded; (3) Samples were distinguished according to the barcode and primers, and the sequence direction was adjusted, exact barcodes were matched, and two-nucleotide mismatches in primer matching were assessed.

Operational taxonomic units (OTUs) with a 97% similarity cutoff were clustered using UPARSE (version 7.1, http://drive5.com/uparse/ (accessed on 5 August 2022)), and chimeric sequences were identified and removed. The taxonomy of each OTU representative sequence was analyzed using the RDP Classifier (http://rdp.cme.msu.edu/ (accessed on 7 August 2022)) against the 16S rRNA database (e.g., Silva SSU138) using a confidence threshold of 0.7. The OTU sequence was flattened (flattening sequence value of 25,946), and the subsequent analysis was carried out with the flattened results [22,23].

### 2.6. Statistical Analysis

The experimental results are shown as the mean ± standard error of mean (SEM). Data were analyzed by GraphPad Prism 7.0 (GraphPad Software, San Diego, CA, USA), and one-way ANOVA was used to determine the differences between groups. Values were considered as significant differences when *p* < 0.05.

## 3. Results

### 3.1. Variation in Environmental Control Variables and PM2.5 Concentrations

The environmental control variables of the whole rearing cycle in broiler houses are shown in Figure 2. The temperature was controlled to decrease slowly with day-age to meet the needs of the broilers. The relative humidity was constantly changing, similar to the outdoor environmental variation basically. The variation tendencies of CO_2_ and NH_3_ were relatively similar in the middle and late stages, showing that they were kept at low concentrations after violent reductions. There were relatively sharp declines in CO_2_ and NH_3_ concentrations (Figure 2B) in the stages of D8 to D10, D32 to D36, and D39 to D41 (Figure 2C,D). The presence of NH_3_ could not even be detected after D28. PM2.5 concentrations indoors increased in the early stage and declined gradually after D28, at which the ventilation of the house reached the maximum (Figure 2E). The wind speed was below 0.5 m/s before 21 day-age, and then strengthened at D21–24 (0.8 m/s) and D24–28 (1.3 m/s). The PM2.5 was 362.68 ± 29.02 μg/m^3^ at D28 and reduced to 142.23 ± 10.78 μg/m^3^ at D42.

### 3.2. Variation in Airborne Bacterial Community Structure of PM2.5 throughout the Rearing Cycle

In total, 12 indoor and outdoor PM2.5 samples were selected for bacterial sequencing. Compared with the database, D10, D24, and D38 were clustered to obtain 229, 290, 236 OTUs, respectively. The Chao indices were 473.80 ± 16.16, 661.22 ± 14.31, and 620.55 ± 29.56 in D10, D24, and D38, respectively. The Shannon indices in D10, D24, and D38 were 3.14 ± 0.21, 4.21 ± 0.05, 4.01 ± 0.09, respectively (Figure 3A,B). The highest alpha diversity (Chao index and Shannon index) was found in D24, which exhibited a significant difference from D10 (*p* < 0.05). The alpha diversity in D38 was also higher than D10, but not significantly (*p* > 0.05).

Principal coordinate analysis (PCoA) indicated that the airborne bacterial community structure exhibited significant differences (R^2^ = 0.6958, *p* = 0.001) between each stage (Figure 3C). The results showed that the weighted values of the PC1 and PC2 axes explained 64.07% and 25.49% of the difference, respectively. The D10 group differed from both D24 and D38 on the PC1 and PC2 axes, and D24 showed a difference from the D38 group on the PC2 axis.

According to the annotation information of each OTU and the distribution and expression of OTU in different samples, we calculated each sample taxonomically at both the phylum and genus level (Figure 3D and Appendix A). In total, 24 phyla, 56 classes, 133 orders, 236 families, and 556 genera were identified in all samples. Inside the house, the bacterial community structures on D10, D24, and D38 exhibited high similarity at the phylum level. The phyla *Firmicutes*, *Proteobacteria*, *Bacteroidota*, and *Actinobacteriota* accounted for more than 99%. In total, 24 genera were found with a relative abundance greater than 1%. The main taxa were *Macrococcus*, *Lactobacillus*, *Ruminococcus_torques_group*, *Rothia*, an unidentified genus of the family *Lachnospiraceae*, *Blautia*, *Enterococcus*, *Corynebacterium*, *Faecalibacterium*, and *Sellimonas*. In these taxa, the most abundant genera were *Macrococcus* (32.9%) in D10, and *Lactobacillus* in D24 (13.6%) and D38 (22.0%). The compositions of outdoor bacteria were similar at the phylum level. Compared with the indoor atmosphere, the numbers of outdoor bacteria genera with a relative abundance greater than 1% were greater, and the compositions were also very different at all rearing stages (Figure 3D).

Genera with a relative abundance greater than 1% were selected for subsequent analysis based on the species annotation information and abundance information. Ternary diagram analysis revealed the distribution and variation in dominant genera in airborne bacterial community (Appendix A). At genus level, the dominant positions were occupied by *Enterococcus*, *Aerococcus*, *Macrococcus*, and *Streptococcus* in D10, which changed to *Faecalibacterium*, *CHKCI001*, and *Christensenellaceae_R-7_group* in D24, then to *Corynebacterium*, *Lactobacillus*, and *norank_f__norank_o__Clostridia_UCG-014* in D38.

*Enterococcus*, *Aerococcus,* and *Macrococcus* were significantly enriched at D10 in the LEfSe analysis (LDA threshold was 2.0, Appendix A). The *Faecalibacterium*, *CHKCI001* (dominant genera), and *Sellimonas* were significantly enriched at D24. In D38 samples, the significantly enriched genera were *Eubacterium_hallii_group*, *Christensenellaceae_R-7_group*, and *Lactobacillus*.

### 3.3. Effects of Environmental Control Variables on the Bacterial Community Structure in PM2.5 Samples

Correlation analysis showed that the genera (greater than 1%) of *Lactobacillus*, *Corynebacterium*, *Eubacterium_hallii_group*, *Christensenellaceae_R-7_group*, and *Micrococcus* were positively and significantly correlated with relative humidity and wind speed, and exhibited a negative correlation with CO_2_, NH_3_, temperature, and PM2.5 (Figure 4A). Other genera, *Enterococcus*, *Streptococcus*, and *Kurthia*, which were greatly affected by environmental aspects, exhibited significant positive correlations with CO_2_, NH_3_, temperature, and PM2.5 (*p* < 0.05), and a significant negative correlation with relative humidity and wind speed (*p* < 0.05). Distance-based redundancy analysis (db-RDA) showed that the first two components (CAP1 and CAP2) together explained 63.30% of the total variation in community structure (Figure 4B). In these environmental control variables, temperature, CO_2_, NH_3_, and PM2.5 concentrations showed inverse correlations with relative humidity and wind speed. The community distribution of D10 was positively correlated with temperature, and the *Enterococcus* and *Macrococcus* were significantly enriched. Relative humidity and wind speed had a considerable impact on the bacterial community structure of the D38 group, with *Lactobacillus* and *Corynebacterium* being significantly enriched (*p* < 0.05, Appendix A). These two groups were contrastingly affected by environmental control variables, whereas the D24 group exhibited a very weak correlation with the environmental control variables. Linear ranking regression analysis was used to evaluate the relationship between alpha diversity and environmental control variables (Table 1). The temperature significantly affected the Shannon index (R^2^ = 0.4878, *p* < 0.05). The NH_3_, CO_2_, and wind speed had a certain influence on the Shannon index, but did not reach the significance level. In conclusion, temperature was the predominant factor in shaping the bacterial community structure carried by PM2.5 in broiler house.

### 3.4. Comparative Analysis of Bacterial Community in Feces and PM2.5

Each cryopreservation tube of D10, D24, and D38 manurial sample was divided into four units for bacterial sequencing. The manurial bacterial community composition revealed that Firmicutes accounted for an absolute proportion at the phylum level, and the rest were *Proteobacteria*, *Bacteroidota*, and *Actinobacteriota*, in order of abundance (Figure 5A and Appendix A). At the genus level, the compositions of the three stages differed greatly. The dominant genera in D10 were *Lactobacillus* and *Enterococcus*, and the genus in D24 and D38 was *Lactobacillus*. As shown in Figure 5B, the D24 manurial samples were clustered closely together. In contrast, the D38 manurial samples were distributed dispersedly with longer distances. There was also a clear structural separation between bacteria in the manurial group and airborne bacteria in the house (R^2^ = 7170, *p* = 0.001).

The Venn diagram in the PM2.5 samples showed that the D10, D24 and D38 groups contained 229, 290, and 236 genera, respectively. The numbers of bacterial genera identified in manure were 197, 125 and 154, respectively, in the three stages (Figure 5C). Inside the house, there were 153 (66.81%), 110 (37.93%), and 129 (54.66%) genera in D10, D24, and D38 shared with manure and 76 (33.19%), 180 (62.07%), and 107 (45.34%) unique bacterial genera found in D10, D24, and D38, respectively. Among the common bacteria, *Enterococcus* (19.83%), *Lactobacillus* (15.94%), *Macrococcus* (14.17%), and *Escherichia-Shigella* (9.42%) in D10, and *Lactobacillus* in D24 (59.73%) and D38 (32.91%) accounted for large proportions; these were all manure-origin bacteria.

### 3.5. Potential Pathogens in Airborne PM2.5

Notably, according to the “List of Pathogenic Microorganisms infected with Humans” issued by the Ministry of Health of the PRC, inside the house we detected the following eight potentially pathogenic genera in each stage: *Enterococcus*, *Corynebacterium*, *Staphylococcus*, *Streptococcus*, *Acinetobacter*, *Escherichia-Shigella*, *Bacteroides*, and *Pseudomonas* (Table 2). In our current study, the proportion of total harmful bacteria in D24 was significantly lower than that in D10 and D38 (*p* < 0.05). *Enterococcus* was detected in all samples and had a relative abundance as high as 9.56% during D10. Similarly, the abundance level of *Corynebacterium* was higher in D38 (9.94%) than other stages (*p* < 0.05). In addition, the proportion of *Streptococcus* in D10 was significantly higher than that in other stages (*p* < 0.05). The effects of potentially pathogenic bacteria affected by environmental control variables are shown in Figure 6. The genera *Enterococcus*, *Corynebacterium*, and *Streptococcus* with higher levels of abundance were significantly affected by environmental control variables (*p* < 0.05), whereas the others were slightly affected (*p* > 0.05).

## 4. Discussion

Livestock and poultry farms emit high concentrations of PM2.5. Due to its larger specific surface area, PM2.5 can carry more airborne bacteria, including some pathogenic bacteria. These emissions lead to environmental pollution around farms, with negative impacts on animal and human health [24]. These emitted bacteria can cause airway inflammation, allergic reactions, disease development, and infection [25,26]. The accumulation of bacterial aerosols, which poses a threat to the health of poultry and producers, is exacerbated by intensive broiler farming methods [18]. This study compared the airborne bacterial communities in PM2.5 between different stages of intensive broiler rearing, to provide a feasible reference for the control of bacterial aerosols and harmful bacteria in broiler houses.

In this study, temperature varied with day-age from 24.2 °C to 33.8 °C and was consistently maintained within the optimal range for broilers. The indoor relative humidity was affected by outdoor atmosphere to a certain extent and fluctuated in the range of 40–60% to ensure the comfort of broilers [1]. Indoor CO_2_ and NH_3_ concentration changes were consistent, from high concentration levels in the early stage and decreasing to stable low concentrations in the middle and late stages. Zhao et al. detected NH_3_ in traditional cage houses with values of 2.8 to 6.7 ppm [27]. Manure is considered to be the main source of indoor NH_3_. Costa et al. detected NH_3_ at levels of 5.37 ppm (open manure storage), 4.95 ppm (removal by manure belt), and 3.85 ppm (litter and removal by belt) in traditional cages of different manure treatments [28]. NH_3_ concentrations of 12–25 ppm were previously detected in a broiler house in summer, but after the replacement of ventilation facilities, NH_3_ was reduced to 2.8–6.7 ppm [29]. Therefore, the concentration of NH_3_ inside the house was highly correlated with the manure cleaning frequency and strategy. In the early stage of rearing, the frequency of manure cleaning and ventilation intensity were kept at low levels, resulting in the accumulation of CO_2_, NH_3_, and PM [30]. As the temperature requirement of broilers decreased and the ventilation volume increased, ammonia, carbon dioxide, and PM all decreased significantly in the late stage.

The cultivable bacteria in the air only accounted for less than 10% of the total; if the cultivation method is adopted, the relevant analysis of bacteria in the air will be greatly limited [31]. The current commonly used method is 16S rRNA high-throughput sequencing to replace the traditional culture method to study the bacterial community structure of airborne bacteria [32]. The environmental parameter changes in the selected control group were completely random and were not affected by the changes in indoor environment and broiler age. In our study on the air in intensive cage-houses, the bacteria community structures of PM2.5 were significantly different from the outdoor atmosphere. A rigorous and systematic cleaning and disinfection process was carried out for one week before a new rearing cycle began. The α-diversity indicated that the abundance and diversity of bacterial aerosols were low in the early stage, then increased rapidly, and remained relatively stable in the middle and late stage. The extremely high Chao index in outdoor atmospheric aerosols indicated that there were more bacterial species [33]. Similar results have been found in other studies, which may be due to the diverse sources of microbes in the atmosphere, which is also closely related to the climatic characteristics of the experimental sites [34]. From the results of the PCoA, it can be seen that there were also significant differences in the bacterial structure between outdoor atmosphere and cage-house environment. In terms of the community composition, even if there were many identical bacteria, their proportions inside and outside the house varied significantly. Khan et al. also reported a similar conclusion [10]. In addition, the unique populations of broiler house differed considerably between the growth stages. The Chao and Shannon indexes reached the highest in the middle growth stage. Similarly, Jiang et al. achieved the same result: the indexes of Chao1, Simpson, and ACE reached the maximum in the middle growth stage of broilers [13].

The PCoA results showed that there were also significant differences between the indoor early stage and middle–late stages. As the dominant phylum, *Firmicutes* consistently comprised the vast majority of bacterial communities during the rearing cycle. The dominant genera in early stage were *Macrococcus* and *Enterococcus*, whereas *Lactobacillus* dominated in the middle–late stages. The LEfSe hierarchical structure further explained which bacteria contributed to the differences in community composition. It was shown that *Enterococcus*, *Macrococcus* (D10), *Sellimonas*, *Faecalibacterium* (D24), and *Lactobacillus* (D38) were critical genera for the stage-specific differences in community. The presence of these differential bacteria led to the separation of the community structure, which was consistent with the beta diversity analysis. In some recent reports, Dai et al. found that *Staphylococcus*, *Faecalibacterium*, *Ruminococcus*, *Corynebacterium*, and *Blautia* accounted for a large proportion in the analysis of the microbial community of the chicken house [1], whereas Wu et al. reported that *Jeotgalicoccus*, *Facklamia*, *Psychrobacter*, *Brachybacterium*, and *Brevibacterium* accounted for dominance [12]. Although the results of the dominant taxa were not completely consistent, several related studies have shown that the bacterial community structure in broiler houses is determined by environmental factors and is considerably different from that in the atmosphere [13,35]. Our results on significant differences in airborne bacteria revealed that the variation in bacterial characteristics is large and random in atmospheric environments without human intervention, whereas changes in the abundance, diversity, composition, and structure of bacteria in a broiler house environment are mainly affected by artificial controls.

The characteristics of airborne bacterial communities in enclosed housing are mainly influenced by environmental control variables. The bacterial communities from different stages were significantly different from each other, because they did not cluster together in the RDA (Figure 4B). The differences in bacterial community structure were shaped under the influence of periodic environmental control variations in the whole rearing cycle. For the intensive rearing of broilers in cages, heating and ventilation were the two artificial controls with contrasting effects, to achieve accurate indoor temperature. Temperature, generally regarded as the most important environmental factor, not only promoted the release and growth of bacteria, but also affected the suspension and diffusion of bacteria to a certain extent. Higher temperatures provide more favorable conditions for the survival and reproduction of bacteria, whereas lower temperatures are not suitable for bacterial growth [36]. Many studies have found that there is a significant correlation between airborne bacteria and temperature and relative humidity. The airborne bacteria concentrations were positively correlated with temperature and were negatively correlated with relative humidity [37,38,39]. In the process of broiler rearing, the houses were maintained at high temperature and with limited ventilation in the early stage. The relative abundance of *Enterococcus* and *Macrococcus* received the benefits of high temperature in the early stage and decreased in the late stage. In contrast, as the day-age of broilers increased, ventilation was enhanced to promote heat dissipation. The contents of *Lactobacillus* and *Corynebacterium* gradually increased in the middle and late stages, which were significantly negatively correlated with temperature and the concentration of NH_3_, CO_2_, and PM2.5. This indicates that the airborne bacterial community changed greatly during the whole broiler rearing cycle under the direct influence of temperature and wind speed, which is consistent with the facts of temperature and ventilation control. Consequently, airborne bacterial community characteristics are shaped by the artificial environment in intensive broiler houses. Temperature and ventilation controls constitute the predominant factors driving the shift in bacterial community structure.

The emissions of manurial bacteria have made manure one of the key sources of airborne bacteria animal houses. Studies have shown that the main factor affecting the bacterial community is the animal species in both manure and aerosol samples. The community structure and the dominant bacteria were different between manure and bioaerosol samples. In livestock or poultry rearing, the community structures and dominant bacteria of manurial and bioaerosol samples are significantly different, due to the hugely different conditions between the aerosol and intestinal environments [37]. Our study found that some genera with higher abundance would be dominant in manure, which may escape into the air and eventually influence the airborne bacterial communities in livestock and poultry houses. These common high-abundance genera included *Bacteroides*, *Lactobacillus*, *Enterococcus*, *Corynebacterium*, *Faecalibacterium*, etc. Bacterial succession also exists throughout the lives of poultry and livestock [39,40,41,42]. The *Christensenellaceae_R-7_group*, one member of the *Firmicutes* phylum, was isolated and cultured from manure [43]. *Enterococcus* is an important member of the gut microbiota, often found in the gut and manure of humans, poultry, livestock, and wild animals [44]. *Corynebacterium*, which is mainly parasitized in animal and human guts, had previously been reported as a common genus in airborne species [6]. The *Eubacterium_hallii_group* is an anaerobic, Gram-positive bacterium frequently found in mouse and human manure [45]. *Macrococcus* exists widely in nature as an animal symbiotic and is one of the most frequently detected bacteria in the air and manure [46]. *Faecalibacterium*, which is present in agricultural species such as dairy cows, pigs, and poultry, is one of the most abundant and important symbiotic bacteria in the animal gut microbiota [47,48]. Key manurial-related bacteria such as *Macrococcus*, *Lactobacillus*, *Enterococcus*, *Corynebacterium*, *Faecalibacterium*, *Eubacterium_hallii_group*, *Christensenellaceae_R-7_group*, *Streptococcus*, and *Escherichia-Shigella* were found in all air samples in this study [49]. We further compared airborne and manure-origin bacteria and found that they shared a large number of dominant bacteria despite their different community structures. The Venn diagram (Figure 5C) showed that the number of unique genera in manure was much lower than the common and unique samples from indoor aerosols. This means that the vast majority of manurial bacterial taxa can be found in aerosols. Furthermore, the common parts in the aerosols and manure were consistent with the main dominant genera, *Enterococcus*, *Lactobacillus*, *Escherichia-Shigella*, and *Macrococcus*, which are widely reported as typical genera [13,50]. This may further suggest that manure is an important source of airborne bacteria inside poultry houses.

Potentially pathogenic bacteria genera carried by PM2.5 in poultry house were closely related to human and animal health. Existing studies have shown that the death rate of chickens is associated with the total number of bacteria; the higher the total number of bacteria, the higher the chicken death rate [51]. In this study, we identified potentially pathogenic bacteria, such as *Enterococcus*, *Corynebacterium*, *Staphylococcus*, *Streptococcus*, *Acinetobacter*, *Escherichia-Shigella*, *Bacteroides*, and *Pseudomonas*, which occupied a large proportion. The pathogenic strains of *Enterococcus*, a group of lactic acid bacteria that occur naturally in the intestine, are an important cause of bone disease in broilers and broiler breeders [52]. *Corynebacterium*, an opportunistic pathogen in immunocompromised hosts, can cause severe pharyngitis and tonsillitis, endocarditis, septic arthritis, and osteomyelitis, and is becoming a major global health problem [53]. *Acinetobacter* can cause respiratory and lung infections, septicemia, meningitis, and other diseases in animals [54]. PM2.5 can carry conditional potentially pathogenic bacteria such as *Escherichia coli* and cause pollution on large-scale farms. *Escherichia coli* can be discharged into the air through breathing and manure by chickens and spread by adhering to PM2.5. The result may be colibacillosis and gastroenteritis when *Escherichia coli* spreads in large quantities inside broiler houses, which greatly increases the mortality of chickens [55]. Kumari et al. found that potentially pathogenic bacteria were widely derived from manure and dander, and were also affected by a variety of environmental control variables [56]. In our research, we found that *Enterococcus*, *Corynebacterium*, and *Streptococcus* were mostly affected by temperature, relative humidity, and ventilation. *Enterococcus*, which is adapted to high temperature, accounted for the largest proportion of PM2.5 in the early stage; subsequently, the proportion decreased rapidly. *Streptococcus* also exhibited a similar variation. The low temperature and high relative humidity environment in the late stage was conducive to the survival of *Corynebacterium*. The ventilation rate was enhanced in the late stage, but many potentially pathogenic bacteria, such as *Corynebacterium*, were still found. Many potentially pathogenic bacteria can be released into the atmospheric environment, becoming a serious pollution source and threatening human health. Environmental control variables have an important influence on the characteristics of potentially pathogenic bacteria; thus, it is necessary to verify the influencing mechanisms in detail.

## 5. Conclusions

As a broiler rearing model which has emerged in China in recent years, cage housing systems can greatly improve the air quality of houses compared with traditional practices. Especially in the middle and late periods of broiler development, ammonia, carbon dioxide, and particulate matter concentrations can be controlled at relatively low levels under conditions where broiler body size and activity are increased. Based on this research, we can draw the following conclusions: (1) Environmental control variables are key factors which cause regular changes in airborne bacterial communities in broiler houses. Temperature and wind speed variations significantly affected the diversity of bacterial communities by changing the dominant taxa at different stages; (2) Manure is an important source of airborne bacteria inside broiler houses; (3) In the middle and late stages, the bacterial communities tend to stabilize, although the populations of potentially pathogenic bacteria grow rapidly and may cause disease. Therefore, attention should be paid to the emission of bacterial aerosols, and preventive measures should be taken. Moreover, future discussions should focus on the physiological activity of bacteria and establish effective biosafety monitoring and disinfection processes in broiler houses to safeguard both animal and human health.

## Figures and Tables

**Figure 1 ijerph-20-00723-f001:**
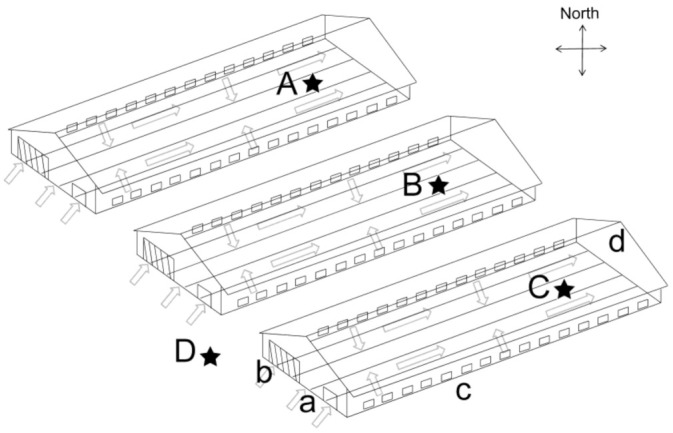
Schematic diagram of the broiler houses’ internal structures. The arrows represent directions of air flow. The sampling positions were at the same height as the middle-layer cages, located at A, B, C, and D (outside of cooling pad). a, entrance; b, evaporative cooling pad; c, variable side inlets; d, exhaust fans.

**Figure 2 ijerph-20-00723-f002:**
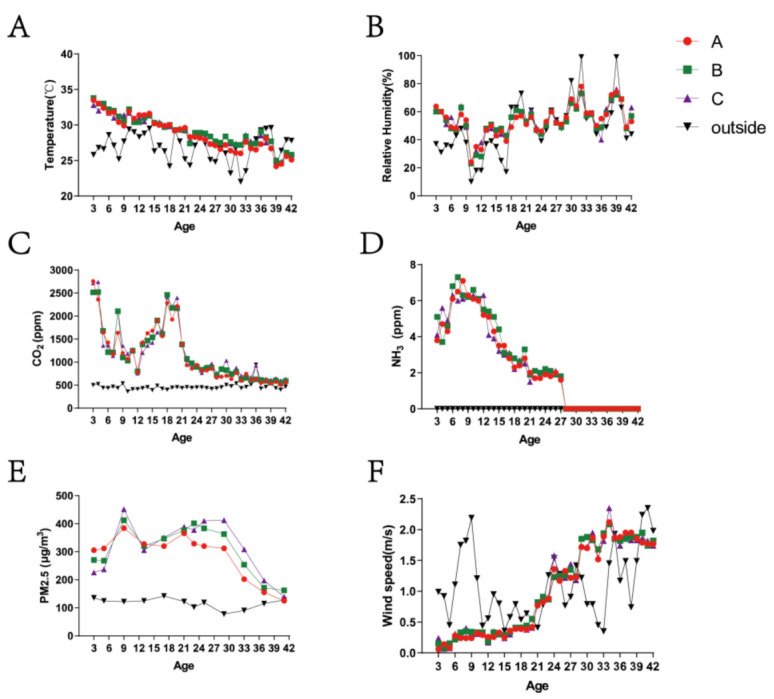
Variations in environmental control variables and PM2.5 concentrations inside and outside the cage-house. (**A**) Temperature (°C); (**B**) relative humidity (%); (**C**) CO_2_ (ppm); (**D**) NH_3_ (ppm); (**E**) PM2.5 (ug/m^3^); (**F**) wind speed (m/s). Legend: A, B, and C in the upper right corner represent the adoption points of three broiler houses.

**Figure 3 ijerph-20-00723-f003:**
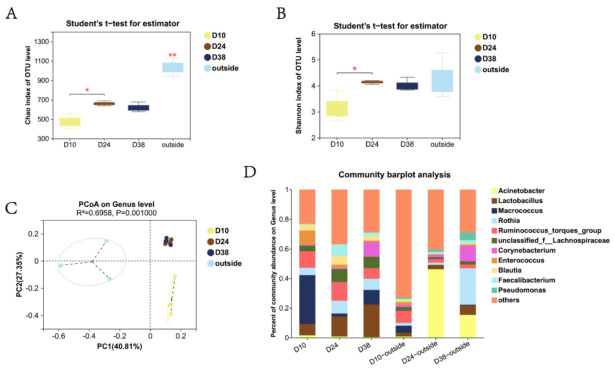
Characteristics of airborne bacterial communities inside and outside the house. (**A**,**B**) represent the alpha diversity of the community: (**A**) Chao index; (**B**) Shannon index; (**C**) PCoA analysis of community structure; (**D**) relative abundance at genus level.* *p* < 0.05, ** *p* < 0.01.

**Figure 4 ijerph-20-00723-f004:**
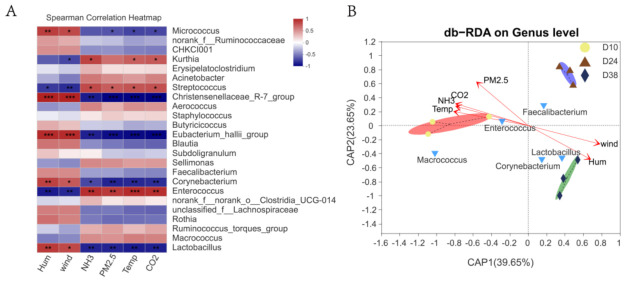
Correlation analysis of environmental control variables and the bacterial community of PM2.5: (**A**) Spearman correlation heatmap of bacteria and environmental control variables at the genus level; (**B**) db-RDA between environmental control variables and bacterial communities. * represents *p* < 0.05; ** represents *p* < 0.01; *** represents *p* < 0.001.

**Figure 5 ijerph-20-00723-f005:**
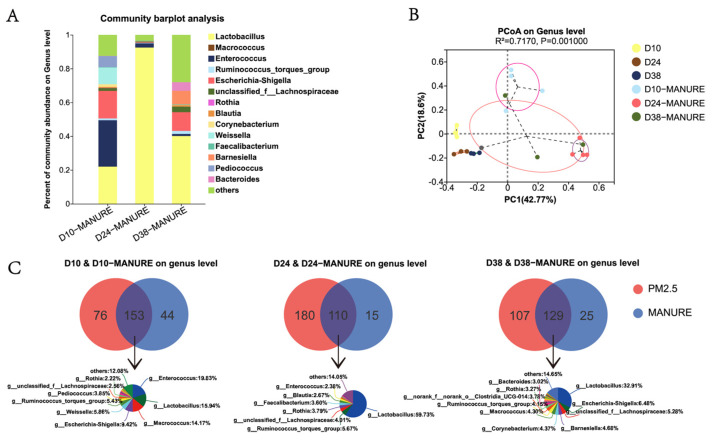
Manurial and indoor airborne bacterial community characteristics: (**A**) the relative abundance of manure at the genus level; (**B**) PCoA of community structure; (**C**) Venn diagram showing the distribution of the same and unique bacteria in feces and PM2.5 at different stages.

**Figure 6 ijerph-20-00723-f006:**
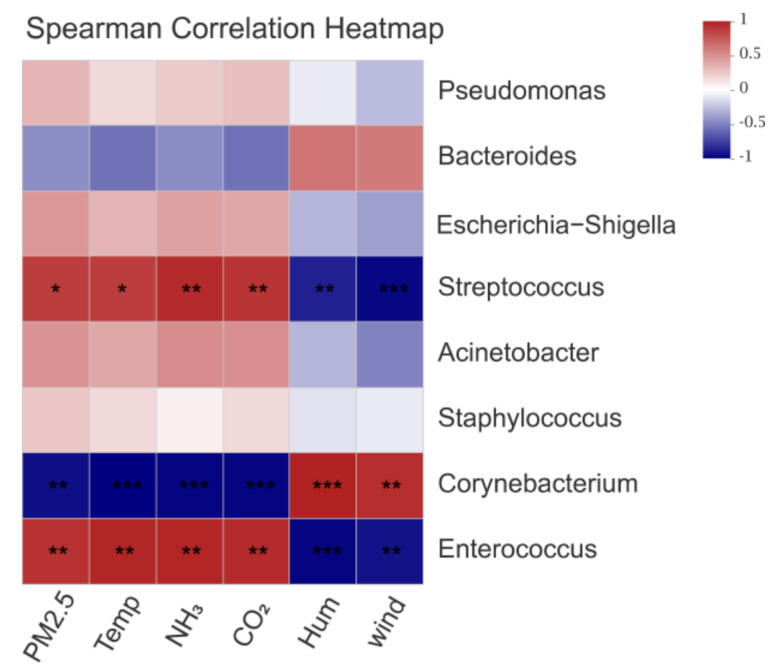
Spearman correlation analysis heatmap of potentially pathogenic bacteria with potential pathogens and environmental control variables. * represents *p* < 0.05; ** represents *p* < 0.01; *** represents *p* < 0.001.

**Table 1 ijerph-20-00723-t001:** Linear ranking regression of environmental control variables and α-diversity.

		PM2.5	Temperature	Humidity	NH_3_	CO_2_	Wind
Chao	R^2^	0.0048	0.1759	0.0080	0.1078	0.0960	0.0964
	*p*-value	0.8595	0.2611	0.8190	0.3882	0.4173	0.4161
Shannon	R^2^	0.1105	0.4878	0.2299	0.3592	0.4019	0.3729
	*p*-value	0.3821	0.0364 *	0.1916	0.0881	0.0667	0.0807

* represents *p* < 0.05.

**Table 2 ijerph-20-00723-t002:** Proportion of potential pathogens in PM2.5 samples.

	D10	D24	D38	*p*-Value
*Enterococcus*	9.56 ± 1.94 ^a^	2.17 ± 0.11 ^b^	1.02 ± 0.08 ^b^	0.003
*Corynebacterium*	0.67 ± 0.27 ^b^	0.87 ± 0.02 ^b^	9.94 ± 2.56 ^a^	0.007
*Staphylococcus*	1.23 ± 0.01	1.51 ± 0.14	1.00 ± 0.28	0.221
*Streptococcus*	2.30 ± 0.79 ^a^	0.34 ± 0.10 ^b^	0.31 ± 0.07 ^b^	0.035
*Acinetobacter*	1.60 ± 0.62	0.82 ± 0.31	0.41 ± 0.05	0.188
*Escherichia-Shigella*	0.18 ± 0.04 ^a^	0.26 ± 0.02 ^a^	0.07 ± 0.01 ^b^	0.007
*Bacteroides*	- ^b^	0.30 ± 0.10 ^a^	0.14 ± 0.06 ^a^	0.052
*Pseudomonas*	0.12 ± 0.05	0.10 ± 0.03	0.03 ± 0	0.208
Total	15.68 ± 0.68 ^a^	6.37 ± 0.49 ^b^	12.92 ± 2.76 ^a^	0.019

^a^ and ^b^, significance at the 0.05 level; -, abundance < 0.1%.

## Data Availability

The sequencing data have been uploaded to the NCBI SRA database under accession number PRJNA845019. The SRA submission number is SUB11561120.

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
