# Peer review of "Bacterial Community Characteristics Shaped by Artificial Environmental PM2.5 Control in Intensive Broiler Houses"

_ijerph, 2022, doi:10.3390/ijerph20010723_

Round 1
Reviewer 1 Report
In this study, the characteristics and influencing factors of bacterial communities in PM2.5 of broiler cage-houses were investigated. This study could provide useful information in regard to airborne bacteria in intensive broiler farming. There are some issues that need to be addressed.
(1) Section 2.2: how many samples were collected, when and where? Please show the details for a better understanding.
(2) Section 3.1 and elsewhere: the environmental control variables, did the authors mean environmental variables? If not, how to control the environmental variables during the experiment? I think the authors just recorded the variation of environmental variables, not controlled. If so, please delete the “control” throughout the manuscript.
(3) Line 181-184: the variation tendencies of CO2 and NH3 were different judging from Fig.2C and D. These sentences should be rephrased.
(4) Section 3.2 and Fig. 3: I think the samples of Outside were collected at D10, D24 and D38, and only one sample at each time. Based on Fig. 3D, the composition and structure of bacterial community from these 3 samples were clearly different. Thus, they could not be classified into one group. So, Fig. 3A, B and C should be revised, and the related contents should be rephrased.
(5) Both Fig. 3D and Fig. 5A showed the percent of community abundance on genus level. Did the D10, D24 and D38 mean the same samples? What was the difference? It is better not to use the same data twice in different figures.
(6) Fig. 6B: there are not related contents describing this figure. So, why use it?
Reviewer 2 Report
The manuscript entitled "Bacterial community characteristics shaped by artificial environment control in PM2.5 from intensive broiler houses" submitted to International Journal of Environmental Research and Public Health by Wang and co-workers is an interesting study on phased changes and influencing factors of airborne bacteria community in broiler houses. They used 16S rRNA gene sequencing to identify the variation of airborne bacterial community structure of PM2.5 samples. The authors found that airborne bacterial communities changed with growth stages, they were influenced by house environment control variables, and feces were one of the main sources. However, there are some suggestions for improvement.
My concerns about this manuscript include:
1. The first and last paragraphs of the Introduction section explain the purpose of the study and the health hazards of airborne microorganisms, respectively. But others are confusing and need to be reorganized.
2. The section 3.4 with the title of “Effect of Manurial Bacteria on Airborne PM2.5 Bacterial Community”, is essentially a comparative analysis of fecal and airborne bacterial community. The title needs to be reworked, and some relevant conclusions are not rigorous and may need to be revised in the Discussion section.
Small changes are advised below:
1. L33-36, lack of references.
2. L37-39, “In intensive farming, airborne particulate matters were derived from the aerosolization of animal manures, feed stuff, skin and feather fragments, escaping into the air, and forming the bioaerosol with large amounts of bacteria in it.” Please rephrase this sentence.
3. L49, lack of attributive element.
4. L57, change “gram-negative” to “Gram-negative”.
5. “Escherichia coli” and “E. coli”, please keep the uniform format.
6. L106, please change the area unit to the correct format.
7. L127, change “hours” to “h”.
8. L151, modify the format of “ddH2O”.
9. Adds a space between the number and the unit.
10. L208-209, “The highest alpha diversity had been found in D24, in which showed a significant difference to D10 (P<0.05).”. It is not clear, please rephrase it.
11. L212, please check the contents in brackets.
12. L254, check the format of “CO2” and “NH3”
13. Figure 6B is not involved in the text.
14. Check and revise the format of reference.
Reviewer 3 Report
1. In the abstract, the results should be more detailed and should be different from the conclusions. Please rewrite it.
2. line 39-41, “In addition, the of PM2.5 played a vital role in air pollution and pathogenic bacteria transmission …”, wrong sentence.
3. line 45, “Skóra et al. 45 (2016) “, please delete the year, you can write as “Skora et al. [4]”. Please check the full text.
4. line 69, please define the abbreviation of “E. coli”.
5. line 75, “Escherichia coli” changed to “E. coli”.
6. line 102, “1620 m2” changed to “1620 m2”.
7. The experimental design should be more detailed, and the sampling time points and the number of repetitions at each time point should be clarified.
8. figure 2, what does “A,B, C” in the upper right corner stand for should also be explained.
9. figure 5, the description of Figure 5C is missing.
10. table 2, please give the specific P value.
11. Please follow the guideline of the manuscript and double check the format of references.
Round 2
Reviewer 1 Report
This manuscript has been improved.
Reviewer 3 Report
Accept in present form.